# Competence Development Strategies after COVID-19: Using PBL in Translation Courses

**Sandra Ribeiro** *[ID], **Célia Tavares** [ID], **Cristina Lopes** [ID] **and Graça Chorão**

CEOS.PP, ISCAP, Instituto Politécnico do Porto, 4465-004 Porto, Portugal
* Correspondence: sribeiro@iscap.ipp.pt

**Abstract:** The 2019 pandemic had a direct impact on all educational stakeholders. While many teachers and trainers regarded the changes with some scepticism, others embraced the opportunity to integrate technology into their teaching-and-learning methods and resources. As translation trainers, it is essential to follow and understand the translation market. Translators require vast competencies, amongst which is the flexibility to adapt. In translation training, project-based learning (PBL) has been established as an essential teaching-and-learning method, as it has proven to foster the development of essential competencies, since it simulates the translator's work environment. Thus, the need to implement new strategies within a short timeframe reinforced the practice of PBL. PBL reflects the work of a freelance translator, because it places the student at the centre of the learning process. In these situations, student self-regulation becomes essential, as it is necessary to analyse the market/situation/project received and be flexible enough to adapt to the specific context. As of 2018–2019, ISCAP implemented PBL as the main teaching-and-learning method in its Technical Translation courses. At the same time, on these courses, an ongoing qualitative quasi-experimental study on student self-regulated learning (SRL) began. The purpose is to understand student perception of their self-regulation competence and its development, or lack thereof, after using PBL to complete translation assignments. The study presented in this article aims to examine the possible effects an online-PBL approach may have on a student's SRL during the pandemic. Students enrolled in the translation courses voluntarily answered a survey on SRL two times: at the beginning and then at the end of the course. The purpose was to analyse and compare each student's responses before and after using PBL strategies, identifying changes in student perception over a six-month period. Additionally, we compare each group's results over a period of three years, which includes the lockdown. Statistical analysis showed that a higher level of self confidence in autonomous learning was achieved, but a lower level of belief in the importance and usefulness of the course contents was noted. Additionally, the study revealed that, with the exception of time-management, student SRL increased. Results indicate that PBL is a useful simulation of the translation labour market and that it does enhance essential competencies, amongst which is student SRL.

**Keywords:** translation; project-based learning; self-regulated learning; COVID-19; teaching and learning



## 1. Introduction

Technology has transformed education across all levels. This happened due to the exponential growth of the Internet and educational technologies, which allowed deep transformations in the teaching-learning process and paradigm over the years. However, it is undeniable that the COVID-19 Pandemic greatly contributed to this process [1,2], because technology became the answer proposed by the need for emergency teaching strategies, capable of guaranteeing student access to classes. As stated by the Council of Europe, "The need to find an alternative to face-to-face learning has spawned numerous experiments in the use of digital technology for education purposes, which have again led to a number of innovations in the use of existing devices and types of software" [3] (p. 18).

While pushed into emergency remote teaching (ERT), educators very quickly had to embrace disruption and implement changes while complying with the learning objectives established for their courses. Although it is important to understand how ERT was converted into online learning, it is also noteworthy to acknowledge how previous experimental teaching and learning methods, due to COVID-19, became the norm.

This paper will consider the effects that the pandemic had on online project-based learning (PBL), in the specific topic of self-regulated learning (SRL), on Technical Translation courses, in the context of the Porto Accounting and Business School (ISCAP), a higher education institution (HEI) in Portugal. It presents a quantitative quasi-experimental study, focused on PBL and its possible effect on SRL during the lockdown. PBL is being researched, not only as a way to simulate a translator's work environment, but also as a practice which promotes the development of competencies, particularly self-regulation, in translation training.

The study described derives from an experimental study implemented in 2018–2019 at ISCAP, prior to the pandemic, led by Zarouk [4]. The experimental study presented positive self-regulation results in an online PBL approach to technical translation assignments. Thus, the online course design and the SRL survey implemented in Zarouk's study were replicated. ISCAP's ongoing study becomes relevant, given that in March 2020 there was a lockdown due to COVID-9 and teacher-and-student interaction abruptly changed from face-to-face to an online, technology-mediated scenario. The same situation occurred, once again, in the spring of the following school year (2020–2021).

The outline of the paper is as follows: Section 2 provides an overview of the theoretical concepts which underpin our study, namely competence development in translation using PBL strategies and SRL. Section 3 describes the methods and procedures implemented to address our research question: Does online PBL foster the development of essential competencies for future translators, namely self-regulation? Is it possible to establish a correlation between the lockdown and student SRL? Section 4 depicts and discusses our findings. Finally, Section 5 provides conclusions that may be drawn from our study.

## 2. Theoretical Framework

### 2.1. Professional Competencies of Translators

Currently, formal training, such as a university degree, is not always required to enable a person to work as a professional translator. This is true for several countries, Portugal included. However, when a translator recognises him/herself as a professional translator, due to having experience working as a translator, it is important that he/she takes into consideration the need to have the competencies to ensure the quality of a translation service. As a result, extensive research has been carried out over the years addressing the competencies translators should have, often inductive in nature. These scholars and/or translators include, to name a few, Harris and Sherwood [5], Chesterman [6], Shreve [7], Neubert [8], Pym [9], and Albir and Taylor [10].

In addition to these studies, it is also important to address standards and guidelines for translation-service providers (TSP), such as the ISO standard—ISO 17100:2015 (amended in 2017) on Quality Translation Services—which lists the competencies that TSP need to have, in order to guarantee translation quality. This list encompasses translation competence; linguistic and textual competence; competence in research, information acquisition, and processing; cultural competence; technical competence, and domain competence [11]. Furthermore, the European Master's in Translation (EMT) network (a quality label established in partnership with the European Commission and higher education institutions) published a framework for a translator's training and translation competence "with future translation graduate employability firmly in mind" [12] (p. 1). The latest version, published in 2022, "has now become one of the leading reference standards for translator training throughout the European Union and beyond" [12] (p. 2), both in higher education institutions and the industry. The initial versions of the framework needed updates to mirror the aims of European translation programmes. Indeed, translation students need to be prepared for an

increasingly dynamic and highly technological workplace. The EMT framework presents five main areas of competence (language and culture; translation; technology; personal and interpersonal; and service provision). The personal-and-interpersonal competence area, for example, regarded as the area which includes generic, often denominated "soft skills", or 21st century skills, includes a descriptor which states: "Check, review, revise and evaluate their own work and that of others according to standard or work-specific quality objectives and assess the appropriateness of using tools for the work at hand" [12] (p. 8). This skill, among others, clearly encompasses the concept of SRL, an essential competence in the era of digital technology, which was particularly relevant during the educational context influenced by the COVID-19 Pandemic.

In translation training, as in other areas, it is important to remember the competencies professionals will need for the evolving labour market. As Albir states, "translator training cannot ignore new pedagogical models that advocate competence-based training and an integrated approach to teaching, learning and assessment" [10] (p. 257). In order to accomplish that ultimate goal, which is preparing students for a real work setting by developing core competencies, the PBL approach is seen as a means to foster SRL in translation students, before and after COVID-19.

### 2.2. Self-Regulated Learning

SRL is considered by Zimmerman as "self-generated thoughts, feelings, and actions that are planned and cyclically adapted to the attainment of personal goals" [13] (p. 14). Describing self-regulation as "cyclical" means that there is a continuous iterative engagement, using task-specific and metacognitive strategies, while gathering information on their value, to achieve defined goals. Despite the theories and models available in the literature across multiple disciplines, most describe self-regulation as a three-phase process: before (initial expectations), during, and after (self-reflection). Zarouk et al., supported by the works of Pintrich [14] and Zimmerman [15], state that "self-regulated learning (SRL) is an active and iterative process in which learners participate metacognitively, motivationally, and behaviourally in their learning process in pursuit of their goals and the contextual characteristics of their environments" [4] (p. 129). Self-regulation is how one controls their thinking, behaviour, emotion, and motivation to attain their goals, using personal strategies.

For professional translators, self-regulation often exceeds the individual and includes a social and interactive component. Indeed, translation has long ceased to be an individual process. The success of (online) collaborative work relies on the self-regulation abilities and methods that each person brings to the group, whether this is in an educational setting or a professional context. Additionally, it is necessary to consider peer assistance throughout the project and within the group (co-regulation), and finally, it is necessary to take into account shared or collective regulation, which includes communication strategies, the regulation of group motivation, and project-coordination strategies (shared regulation) [16].

Thus, we consider that SRL is possible through PBL because it enables students to assume the responsibility for their own learning and become more actively engaged throughout the entire process. Students set their own objectives, track their own development, and adapt their approaches as necessary, to achieve those objectives. Self-regulation helps students become more independent learners and take ownership of their learning. They can identify their strengths and limitations and also, as mentioned before, develop essential skills and competencies.

### 2.3. Project-Based Learning

Teaching and learning methods are continuously changing, but it is crucial to guarantee that the appropriate strategies are used in order to achieve the intended learning outcomes. Moreover, it is important to align education with the labour market, to allow students to understand what is expected of them. The translation market is growing and changing exponentially [17], in line with technological developments. The translation

process is no longer a single-person activity; it is a complex process, a challenge, that very often needs to be prepared, organized and overseen.

In PBL, ill-structured challenges (using either authentic [18] or simulated projects) are used for the learning process, which, according to Tan [19] is an on-going active, student-centred approach. Michel et al. [20] and Shet et al. [21] also claim that PBL demands that students take control of their learning process, fostering the development of subject-matter expertise, teamwork, critical thinking, and problem-solving skills.

Hence, PBL emerges as a plan to bring an idea or action to life, where students are the main actors. In translator training, students are introduced to translation projects, where they learn by doing, i.e., they organise the translation project from the moment they receive it from the "client" until they deliver the final translated document. This means that the students need to organise their time, assume roles, and distribute tasks in order to meet the client's requirements, just like a translation-service provider (either a translation company or a freelance translator). In sum, in a traditional translation classroom, the use of a PBL approach requires the completion of a number of challenging activities in a predetermined amount of time [22].

Studies on PBL in translation training are not new (see, for example, González and Díaz [23], Li et al. [24], Moghaddas and Khoshsaligheh [25], and Apandi and Afiah [26]).

Our approach is, as mentioned previously, supported by Zarouk's et al. [4] study on the impact of online project-based learning on self-regulation in higher education. The authors describe a study conducted in the school year 2018/2019, prior to the pandemic, and report that all the groups that participated in the study claimed to have increased their motivation for learning. The translation group (students from the Technical Translation course at ISCAP) additionally claimed a high level of satisfaction and usefulness when questioned on PBL. These participants appreciated the design of the course, where they worked in teams and where each member had specific roles and tasks. The online PBL scenario, structured on the institutional platform Moodle, sought to replicate authentic translation scenarios. In this study, the authors also conclude that the translation group significantly improved their self-regulation and collaboration strategies in comparison with the other groups that participated in the study. These findings are consistent with similar results suggesting that well-structured but flexible teaching design can promote students' active learning behaviours and teamwork, as suggested by Zarouk et al., who refer to Shih and Tsai [27], as well as Sakulviriyakitkul [28].

Given this context, and the positive results of the previous study, the teachers involved in the project decided to maintain the course design and, once again, study participant perspectives on SRL using online PBL in the Technical Translation courses at ISCAP. It is within this premise that we propose to address the following research questions, as stated previously:

- Does online PBL foster the development of essential competencies for future translators, namely self-regulation?
- Is it possible to establish a correlation between the lockdown and student SRL?

The methods and procedures implemented to provide answers to these questions are depicted in the section that follows.

### 3. Methods and Procedures

The study presented in this article aims to examine the possible effects an online PBL approach may have on a student's SRL during the pandemic. Students enrolled in the translation courses voluntarily answered a survey on SRL two times: at the beginning and then at the end of the course. The purpose was to analyse and compare each student's responses before and after using PBL strategies, identifying changes in student perceptions over a six-month period. Additionally, we compare each group's results over a period of three years, which includes the lockdown due to COVID-19.

Based on the objectives, a quasi-experiment study [29] following a pre-test–post-test design, using quantitative analysis to measure SRL processes and strategies, was adopted.

### 3.1. Sample and Participants

Given the positive and encouraging results from the previous study [3], which pertained to the school year 2018–2019, the teachers decided to replicate the online-PBL design in the Technical Translation courses at ISCAP. The study presented in this article was conducted between March and June of 2019–2020, of 2020–2021, and of 2021–2022, which correspond to the second semester of the school year. Students enrolled in the bachelor's program Administrative Assistance and Translation at ISCAP have Technical Translation courses in the second semester of the second year. Thus, the sample was non-random, conducted during the semester.

At the beginning of the second semester, in 2019–2020, 91 students had enrolled; in 2020–2021, 82 students had enrolled, and in 2021–2022, 79 students had enrolled on the Technical Translation course on ISCAP's Moodle platform, totalling 252 students. Two teachers participated in the planning and structuring of the PBL course design, and then conducted the course and evaluated the students. The teachers adopted the roles of facilitators and clients throughout the different projects. A detailed description of the online-PBL design implemented has already been published [30].

Participation in the projects was mandatory for all students who opted for continuous assessment. However, student participation in the surveys (pre-test and post-test) was voluntary, and subject to an informed consent. Students enrolled on the course were openly invited to participate in the study by filling in the pre-test and post-test surveys on the first and the last sessions. Given the voluntary nature of the study, a significantly lower number of students than those who had initially enrolled on the course answered the pre- and post-test surveys.

To examine the impact of our approach, as the intention was to compare each student's initial and final SRL survey, only the surveys of the students who filled in both the pre-test and post-test were initially sought to be included in this study. Given the reduced number of participants (16), since most would be automatically excluded from the study because there was no consistency, it was decided that the data should be analysed from various perspectives, as will be described below. Although the reasons behind the lack of participation were not researched, as it is outside the scope of the present study, these will still be considered in the discussion.

### 3.2. Data-Collection Instruments

To collect information on individual students' SRL, we used the same instrument as Zarouk et al., the Self-Regulation Project-Based Learning (SRPBL) survey. The SRPBL is an adapted self-report instrument created by the authors to measure self-regulation in online and blended-learning environments [30], adapted from the original Self-Regulation Survey (SRQ), proposed by Brown et al. [31], and validated for consistency. The SRPBL focuses on three main self-regulation pillars (motivation, self-regulation, and collaboration). The instrument comprises 47 descriptors which participants classified on a five-point Likert scale (1 strongly disagree–5 strongly agree). The descriptors can be grouped into three main constructs: Motivational Beliefs (11 descriptors), Self-Regulation Strategies (27 descriptors) and Collaborative Strategies (9 descriptors).

Data collected from each pre-test and post-test was compared using paired-sample statistical tests, and data from the sectional and the longitudinal sample was analysed through confidence intervals, parametric and non-parametric statistical tests.

## 4. Results and Findings

The study presented focuses on two main research question; the first on whether or not online PBL fosters the development of essential competencies for future translators, namely self-regulation; and second, whether it is possible to establish a correlation between the lockdown and student SRL.

Initially, and as described in Section 3, it was only possible to comparatively analyse 16 participants, despite the number of students enrolled on the Technical Translation

course. These participants were the only ones that answered the pre-test (Q) in March and April 2020 (the beginning of the semester) and the post-test (P) in May and June of 2020 (2019–2020) (the end of the semester). This sample was the only consistent group which enabled us to establish a comparison, as we were able to pair participant responses.

For this sample, questions belonging to the constructs, Motivational Beliefs and Self-Regulating Activities Before Learning, were analysed both individually and in groups. For each student, the average of the scores given in the items belonging to Motivational Beliefs was computed and used as value for this construct. Similarly, the average values of the items belonging to every construct were computed for each student. This is described in Section 4.1. Normality of the data was checked with the Shapiro–Wilk test, and parametric or non-parametric tests were then performed accordingly. To compare the results of the constructs in the pre-test and in the post-test, paired-sample *t*-tests were performed. Afterwards, a further analysis of each item was carried out using the Wilcoxon paired-samples test.

In the second phase, given the lack of consistency in the sample, the study proceeded with a longitudinal analysis of the complete post-test (P) survey for 2019–20, 2020–21 and 2021–22 (see Section 4.2). For this analysis, a sample of 49 students was collected. Descriptive statistics and Shapiro–Wilk normality tests were performed separately, by school year, for three levels of variables:

- Item by item;
- Items grouped in nine constructs;
- Items grouped in three main dimensions.

The reliability of the constructs was analysed using Cronbach's alpha. The dimensions, constructs and items were tested for differences across the years with ANOVA, Kruskal–Wallis tests and Mann–Whitney tests.

### 4.1. Comparative Analysis of the Pre-Test and Post-Test for 2019–2020

For each student, the average of the scores given in the items belonging to each of the two constructs present both in the pre-test and in the post-test was computed. These average values were analysed as variables representative of the constructs.

Shapiro–Wilk tests indicated that the variables follow a normal distribution, so parametric tests could be used (*t*-test for paired samples). From these *t*-tests, $p = 0.723$ and $p = 0.422 > 5\%$ were obtained, thus concluding that there are no statistically significant differences between the means of the constructs in the pre-test and post-test (Table 1).

**Table 1.** Paired-Samples *t*-test for comparing the constructs in the Pre-Test and Post-Test.

| | | Paired-Samples *t*-Test | | | | | | |
|---|---|---|---|---|---|---|---|---|
| | | **Paired Differences** | | | | **T** | **df** | **Significance** |
| | | **Mean** | **Std. Deviat.** | **95% Confidence Interval of the Difference** | | | | **Two-Sided *p*** |
| | | | | **Lower** | **Upper** | | | |
| Pair 1 | Motivational Beliefs (Pre-test)—Motivational Beliefs (post-test) | −0.02500 | 0.27689 | −0.17254 | 0.12254 | −0.361 | 15 | 0.723 |
| Pair 2 | Self-Regulation Strategies Before Learning (pre-test)—Self-Regulation Strategies After Learning (post-test) | 0.10000 | 0.48442 | −0.15813 | 0.35813 | 0.826 | 15 | 0.422 |

A closer analysis of each item in the pre-test and post-test was performed.

The Wilcoxon test is a non-parametric test to analyse whether there are differences in paired samples. In this case, we used it to test differences between the answers given in the pre-test and in the post-test, i.e., before and after the learning period (Table 2). This

non-parametric test was used because the sample was small, and most variables did not follow a normal distribution.

**Table 2.** Wilcoxon Test for comparing items in the Pre-Test and Post-Test.

| Wilcoxon Signed-Ranks Test | Z | Asymp. Sig. (2-Tailed) | Result |
|---|---|---|---|
| Self-motivational beliefs: In online learning, I prefer support material that challenges me so that I can learn new things. | −0.905 [b] | 0.366 | No differences |
| Self-motivational beliefs: If I study appropriately, I can understand the contents of this module. | −1.890 [c] | 0.059 | There are significant differences |
| Self-motivational beliefs: I think I am able to use what I learn in other situations. | −1.414 [c] | 0.157 | No differences |
| Self-motivational beliefs: I believe I can do an excellent work in this module. | 0.000 [d] | 1.000 | No differences |
| Self-motivational beliefs: I am sure I can understand the topics presented in the readings | −2.887 [c] | 0.004 | There are significant differences |
| Self-motivational beliefs: Getting a good grade is the most rewarding thing for me. | −1.000 [c] | 0.317 | No differences |
| Self-motivational beliefs: It is important and useful for me to learn the subject of the module | −2.530 [b] | 0.011 | There are significant differences |
| Self-motivational beliefs: I am very interested in this topic (Subtitling) | −0.816 [b] | 0.414 | No differences |
| Self-motivational beliefs: I expect to succeed in this module. | −1.732 [b] | 0.083 | No differences |
| Self-motivational beliefs: I am sure that I will master the module's competence and subject matters. | −0.277 [c] | 0.782 | No differences |
| Metacognitive activities before learning: I think about what I really need to learn before starting a task. | 0.000 [d] | 1.000 | No differences |
| Metacognitive activities before learning: I set short-term (daily) as well as long-term (weekly) goals for online training. | −0.776 [c] | 0.438 | No differences |
| Metacognitive activities before learning: I set goals to help me manage my study time for this module. | −0.647 [c] | 0.518 | No differences |
| Metacognitive activities before learning: I think of alternative ways to solve a problem and choose the best course of action in this module. | −0.277 [b] | 0.782 | No differences |
| Metacognitive activities before learning: At the beginning of a task, I think about the study strategies I will use. | −2.667 [b] | 0.008 | There are significant differences |

a. Wilcoxon Signed-Ranks Test. b. Based on positive ranks. c. Based on negative ranks. d. The sum of negative ranks equals the sum of positive ranks.

The analysis showed interesting results (at the 5% level of significance) for the participants' perceptions in the following three items of the pre-survey and the post-survey: (1) Q: I am sure I can understand the topics presented in the readings—when asked about the level of confidence as to their autonomous study. The findings show that the participants reported a higher level of self-confidence as to learning autonomously and confidently; (2) Q: It is important and useful for me to learn the subject of the module—the results show a slight decrease as to the perceived importance and usefulness of the contents. This may indicate some dependence on teacher guidance, which was normally the case

in a face-to-face context. In the construct Metacognitive Activities (self-regulation) Before Learning, a significant decrease was noted in (3) Q: At the beginning of a task, I think about the study strategies I will use. The participants regarded planning strategies as irrelevant and unnecessary, which may indicate that previous preparation work had little effect on student' performance.

If a 10% significance was considered, it would still be possible to detect the existence of significant differences in the answers given in the pre-survey and in the post-survey for Q: If I study appropriately, I can understand the contents of this module. This refers to participant motivational beliefs and self-awareness as to the importance of studying in order to properly understand the syllabus content. It is apparent that confidence in the effectiveness of studying increased.

In sum, an analysis of the pre-test and post-test in 2019–2020 revealed significant differences in two constructs: Motivational Beliefs and Self-Regulation Strategies Before Learning.

### 4.2. Longitudinal Analysis of the Post-Test for 2019–2020, 2020–2021, and 2021–2022

Opting for a longitudinal analysis, the verified sample comprises a total of 49 students: 20 enrolled in 2019/20, 24 in 2020/21 and only 5 from 2021/22. The full survey obtained an excellent internal consistency, with a Cronbach's alpha of 0.933. Grouping the questions into three higher level dimensions—Motivational Beliefs, Self-Regulation Strategies, and Collaborative Strategies—also revealed a good reliability, with Cronbach's alpha ranging from 0.833 to 0.891. Reliability statistics were mainly good when the questions were grouped by the nine lower-level constructs (descriptors); the exception was for Time Management (Table 3). This could be due to the reduced number of items in this construct, and also to a misunderstanding of the questions presented.

**Table 3.** Reliability analysis of the survey.

| Dimensions | Constructs | Reliability Statistics | | |
|---|---|---|---|---|
| | | Cronbach's Alpha | Cronbach's Alpha Based on Standardized Items | N of Items |
| Full survey | | 0.933 | 0.938 | 58 |
| Motivational Beliefs | Motivational Beliefs | 0.833 | 0.836 | **10** |
| Self-Regulation Strategies | | 0.891 | 0.898 | **27** |
| | Before learning | 0.803 | 0.808 | 5 |
| | During learning | 0.733 | 0.736 | 4 |
| | After learning | 0.814 | 0.819 | 4 |
| | Time management | 0.422 | 0.442 | 4 |
| | Environment Structuring | 0.743 | 0.750 | 4 |
| | Persistence | 0.799 | 0.805 | 6 |
| Collaborative Strategies | | 0.875 | 0.885 | **21** |
| | Peer Learning | 0.850 | 0.857 | 9 |
| | Help Seeking | 0.919 | 0.934 | 12 |

#### 4.2.1. Longitudinal Analysis of the Three Dimensions

The three main dimensions were analysed in the three school years (2019/20, 2020/21, 2021/22) with three different cohorts. Collaborative strategies follow a normal distribution (Table 4), but the other two dimensions do not have a normal distribution in some of the school years. Therefore, for the first two dimensions, non-parametric tests were implemented, while for the third dimension, we used parametric tests.

**Table 4.** Descriptive statistics and normality tests for the dimensions in the longitudinal sample.

| Dimensions | School Year | N | Mean | Std. Dev. | 95% Confidence Interval for Mean | | Min. | Max. | Shapiro–Wilk | | | Test Result |
|---|---|---|---|---|---|---|---|---|---|---|---|---|
| | | | | | Lower Bound | Upper Bound | | | Statistic | df | Sig. | |
| Motivational Beliefs | 2019/20 | 20 | 4.185 | 0.513 | 3.945 | 4.425 | 3.2 | 5.0 | 0.963 | 20 | 0.610 | Normally distributed |
| | 2020/21 | 24 | 4.408 | 0.376 | 4.249 | 4.567 | 3.7 | 4.9 | 0.897 | 24 | 0.018 | Reject normal dist. for 5% significance |
| | 2021/22 | 5 | 4.400 | 0.235 | 4.109 | 4.691 | 4.0 | 4.6 | 0.813 | 5 | 0.103 | Normally distributed |
| | Total | 49 | 4.316 | 0.435 | 4.191 | 4.441 | 3.2 | 5.0 | 0.947 | 49 | 0.029 | Reject normal dist. for 5% significance |
| Self-Regulation Strategies | 2019/20 | 20 | 3.687 | 0.514 | 3.446 | 3.928 | 2.8 | 5.0 | 0.955 | 20 | 0.454 | Normally distributed |
| | 2020/21 | 24 | 3.997 | 0.397 | 3.829 | 4.165 | 3.3 | 4.6 | 0.929 | 24 | 0.091 | Reject normal dist. for 10% significance |
| | 2021/22 | 5 | 3.778 | 0.334 | 3.363 | 4.193 | 3.3 | 4.2 | 0.922 | 5 | 0.545 | Normally distributed |
| | Total | 49 | 3.848 | 0.460 | 3.716 | 3.980 | 2.8 | 5.0 | 0.981 | 49 | 0.602 | Normally distributed |
| Collaborative Strategies | 2019/20 | 20 | 4.133 | 0.463 | 3.916 | 4.350 | 3.3 | 5.0 | 0.964 | 20 | 0.630 | Normally distributed |
| | 2020/21 | 24 | 4.288 | 0.418 | 4.111 | 4.464 | 3.4 | 4.9 | 0.949 | 24 | 0.257 | Normally distributed |
| | 2021/22 | 5 | 4.371 | 0.309 | 3.987 | 4.756 | 4.0 | 4.8 | 0.972 | 5 | 0.885 | Normally distributed |
| | Total | 49 | 4.233 | 0.429 | 4.110 | 4.357 | 3.3 | 5.0 | 0.966 | 49 | 0.171 | Normally distributed |

Testing for differences across the years (Table 5), significant differences in the mean (and median) were found only in Self-Regulation Strategies, for 10% of significance (Kruskal–Wallis $p = 0.087$). The dimensions Motivational Beliefs and Collaborative Strategies do not show significant differences across the years (Kruskal–Wallis $p = 0.293$, and ANOVA $p = 0.378$, respectively). For 5% significance, the variances of the three dimensions can be considered homogeneous across the years.

**Table 5.** Testing the main dimensions across the school years for differences in variance, mean and median.

| Dimensions | Tests of Homogeneity of Variances | | | ANOVA | | Kruskal–Wallis Test | | |
|---|---|---|---|---|---|---|---|---|
| | Levene Statistic Based on Mean | Sig. | Levene Statistic Based on Median | Sig. | F | Sig. | Kruskal–Wallis H | df | Asymp. Sig. |
| Motivational Beliefs | 2.672 | 0.080 | 2.624 | 0.083 | 1.581 | 0.217 | 2.456 | 2 | 0.293 |
| Self-Regulation Strategies | 1.325 | 0.276 | 0.850 | 0.434 | 2.718 | 0.077 | 4.883 | 2 | 0.087 |
| Collaborative Strategies | 0.830 | 0.443 | 0.403 | 0.671 | 0.993 | 0.378 | 2.274 | 2 | 0.321 |

If we consider the evolution of the results across the years, there is an increase in the mean of the dimension Self-Regulation Strategies (green in Figure 1), followed by a

decrease. This increase in self-regulation strategies from 2019/20 to 2020/21 was found to be significant with the Mann–Whitney test ($p = 0.039$), but the decrease from 2020/21 to 2021/22 was not significant ($p = 0.203$). The dimensions Motivational Beliefs and Collaborative Strategies experienced stable results with a light increase in the mean, which is nonsignificant.

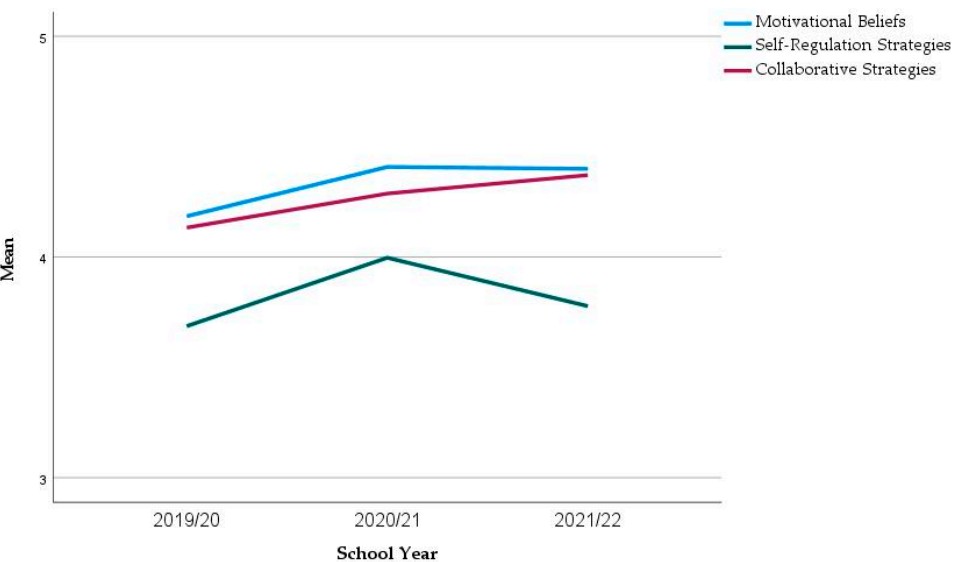

**Figure 1.** Evolution of the mean values in the main dimensions.

### 4.2.2. Analysis of the Constructs

While analysing the nine constructs of the survey in more detail (Table 6), it is possible to see that most of the variables do not follow a normal distribution. Thus, we used nonparametric tests to analyse the differences across the years.

The construct with the highest mean values is Motivational Beliefs, followed by Help Seeking and Peer Learning. The construct with the lowest mean scores is Time Management (Figure 2). The latter may, once again, indicate teacher interference. As this occurred during the lockdown, the need to assure students were keeping up to date may have had a negative effect on student time-management autonomy. On the other hand, it is also possible to see this as a lack of students' time-management skills, in line with the literature [32–35].

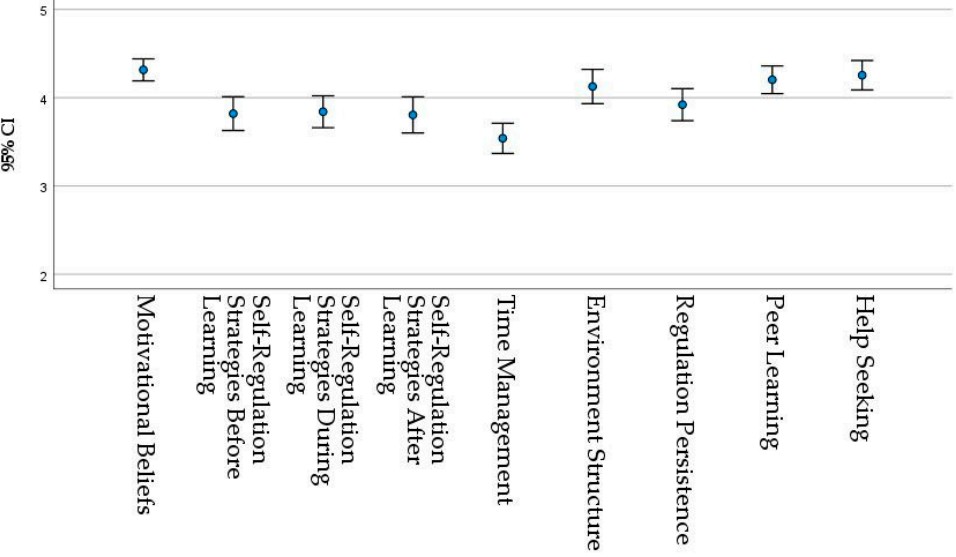

**Figure 2.** 95% confidence intervals for mean of the constructs in the longitudinal sample.

**Table 6.** Descriptive statistics and normality tests for the constructs in the longitudinal sample.

| Construct | School Year | Descriptives | | | | | | | Tests of Normality | | | |
| | | N | Mean | Std. Dev. | 95% Confidence Interval for Mean | | Min. | Max. | Shapiro-Wilk | | | Test Result |
| | | | | | Lower Bound | Upper Bound | | | Statis. | df | Sig. | |
| Motivational Beliefs | 2019/20 | 20 | 4.185 | 0.513 | 3.945 | 4.425 | 3.2 | 5.0 | 0.963 | 20 | 0.610 | |
| | 2020/21 | 24 | 4.408 | 0.376 | 4.249 | 4.567 | 3.7 | 4.9 | 0.897 | 24 | 0.018 | |
| | 2021/22 | 5 | 4.400 | 0.235 | 4.109 | 4.691 | 4.0 | 4.6 | 0.813 | 5 | 0.103 | |
| | Total | 49 | 4.316 | 0.435 | 4.191 | 4.441 | 3.2 | 5.0 | 0.947 | 49 | 0.029 | Reject normal dist. |
| Self-Regulation Strategies Before Learning | 2019/20 | 20 | 3.580 | 0.686 | 3.259 | 3.901 | 2.2 | 5.0 | 0.954 | 20 | 0.433 | |
| | 2020/21 | 24 | 4.092 | 0.472 | 3.892 | 4.291 | 3.2 | 5.0 | 0.969 | 24 | 0.642 | |
| | 2021/22 | 5 | 3.480 | 0.976 | 2.269 | 4.691 | 2.0 | 4.6 | 0.969 | 5 | 0.870 | |
| | Total | 49 | 3.820 | 0.667 | 3.629 | 4.012 | 2.0 | 5.0 | 0.971 | 49 | 0.260 | Normally dist. |
| Self-Regulation Strategies During Learning | 2019/20 | 20 | 3.638 | 0.741 | 3.291 | 3.984 | 2.0 | 5.0 | 0.973 | 20 | 0.819 | |
| | 2020/21 | 24 | 4.052 | 0.500 | 3.841 | 4.263 | 3.0 | 5.0 | 0.966 | 24 | 0.563 | |
| | 2021/22 | 5 | 3.650 | 0.418 | 3.131 | 4.169 | 3.0 | 4.0 | 0.881 | 5 | 0.314 | |
| | Total | 49 | 3.842 | 0.629 | 3.661 | 4.022 | 2.0 | 5.0 | 0.964 | 49 | 0.140 | Normally dist. |
| Self-Regulation Strategies After Learning | 2019/20 | 20 | 3.738 | 0.784 | 3.370 | 4.105 | 1.8 | 5.0 | 0.957 | 20 | 0.478 | |
| | 2020/21 | 24 | 3.938 | 0.618 | 3.677 | 4.198 | 2.5 | 5.0 | 0.941 | 24 | 0.168 | |
| | 2021/22 | 5 | 3.450 | 0.837 | 2.411 | 4.489 | 2.0 | 4.0 | 0.751 | 5 | 0.030 | |
| | Total | 49 | 3.806 | 0.713 | 3.601 | 4.011 | 1.8 | 5.0 | 0.935 | 49 | 0.010 | Reject normal dist. |
| Time Management | 2019/20 | 20 | 3.525 | 0.697 | 3.199 | 3.851 | 2.5 | 5.0 | 0.903 | 20 | 0.046 | |
| | 2020/21 | 24 | 3.531 | 0.485 | 3.326 | 3.736 | 2.8 | 4.8 | 0.939 | 24 | 0.153 | |
| | 2021/22 | 5 | 3.650 | 0.762 | 2.703 | 4.597 | 3.3 | 5.0 | 0.644 | 5 | 0.002 | |
| | Total | 49 | 3.541 | 0.596 | 3.370 | 3.712 | 2.5 | 5.0 | 0.906 | 49 | 0.001 | Reject normal dist. |
| Environment Structure | 2019/20 | 20 | 3.950 | 0.701 | 3.622 | 4.278 | 2.8 | 5.0 | 0.919 | 20 | 0.095 | |
| | 2020/21 | 24 | 4.208 | 0.641 | 3.938 | 4.479 | 3.0 | 5.0 | 0.917 | 24 | 0.051 | |
| | 2021/22 | 5 | 4.450 | 0.671 | 3.617 | 5.283 | 3.5 | 5.0 | 0.852 | 5 | 0.201 | |
| | Total | 49 | 4.128 | 0.675 | 3.934 | 4.322 | 2.8 | 5.0 | 0.928 | 49 | 0.005 | Reject normal dist. |
| Regulation Persistence | 2019/20 | 20 | 3.708 | 0.703 | 3.379 | 4.037 | 2.5 | 5.0 | 0.965 | 20 | 0.656 | |
| | 2020/21 | 24 | 4.090 | 0.590 | 3.841 | 4.339 | 3.0 | 5.0 | 0.944 | 24 | 0.196 | |
| | 2021/22 | 5 | 3.967 | 0.183 | 3.740 | 4.193 | 3.7 | 4.2 | 0.828 | 5 | 0.135 | |
| | Total | 49 | 3.922 | 0.631 | 3.741 | 4.103 | 2.5 | 5.0 | 0.962 | 49 | 0.118 | Normally dist. |
| Peer Learning | 2019/20 | 20 | 4.172 | 0.530 | 3.924 | 4.420 | 3.3 | 5.0 | 0.945 | 20 | 0.291 | |
| | 2020/21 | 24 | 4.273 | 0.510 | 4.058 | 4.488 | 3.4 | 5.0 | 0.928 | 24 | 0.088 | |
| | 2021/22 | 5 | 4.000 | 0.820 | 2.982 | 5.018 | 2.7 | 4.9 | 0.901 | 5 | 0.417 | |
| | Total | 49 | 4.204 | 0.547 | 4.047 | 4.361 | 2.7 | 5.0 | 0.958 | 49 | 0.079 | Reject normal dist. |
| Help Seeking | 2019/20 | 20 | 4.104 | 0.635 | 3.807 | 4.401 | 2.9 | 5.0 | 0.945 | 20 | 0.301 | |
| | 2020/21 | 24 | 4.299 | 0.534 | 4.073 | 4.524 | 3.3 | 5.0 | 0.930 | 24 | 0.095 | |
| | 2021/22 | 5 | 4.650 | 0.410 | 4.141 | 5.159 | 4.0 | 5.0 | 0.871 | 5 | 0.271 | |
| | Total | 49 | 4.255 | 0.580 | 4.088 | 4.422 | 2.9 | 5.0 | 0.937 | 49 | 0.011 | Reject normal dist. |

Figure 3 depicts the evolution of the mean values of the nine constructs throughout the years. Almost all constructs increased after the first year of COVID-19. After two years, three constructs increased their mean values (*Time management*, *Environment Structure*, *Help Seeking*) and the other six constructs decreased. However, through ANOVA and Kruskal-Wallis tests (Table 7), it was possible to find that the only constructs which showed significant differences across the years were *Self-Regulation Strategies (SRS) Before Learning*

and *SRS During Learning* (ANOVA $p = 0.016$ and $p = 0.069$, respectively). A closer inspection of these differences with Tukey's post-hoc test for multiple comparisons proved that there was a significant increase in *SRS Before Learning* and *SRS During Learning* from 2019/20 to 2020/21 ($p = 0.025$ and $p = 0.072$, respectively) but the decrease from 2020/21 to 2021/22 is non-significant ($p = 0.125$ and $p = 0.375$, respectively).

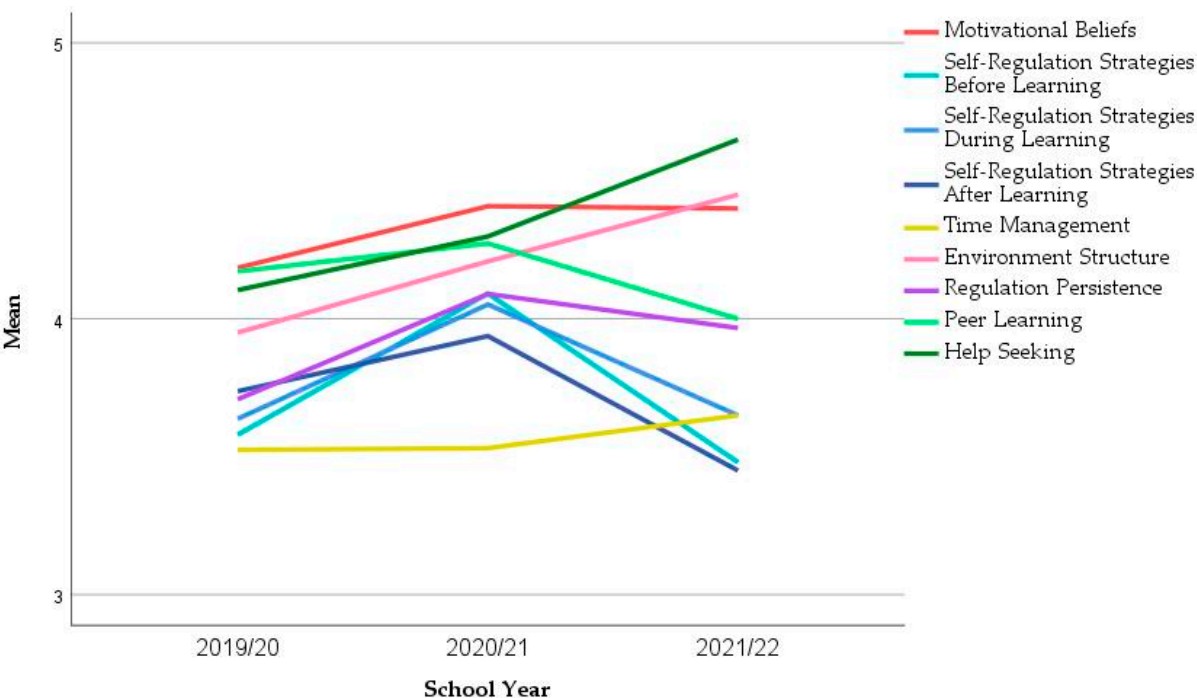

**Figure 3.** Evolution of the mean of the constructs in the longitudinal sample.

**Table 7.** Testing the constructs across the school years for differences in variance, mean and median.

| Construct | Tests of Homogeneity of Variances | | | | ANOVA | | Kruskal–Wallis Test | | |
|---|---|---|---|---|---|---|---|---|---|
| | Levene Statistic based on Mean | Sig. | Levene Statistic based on Median | Sig. | F | Sig. | Kruskal–Wallis H | df | Asymp. Sig. |
| Motivational Beliefs | 2.672 | 0.080 | 2.624 | 0.083 | 1.581 | 0.217 | 2.456 | 2 | 0.293 |
| Self-Regulation Strategies Before Learning | 1.680 | 0.198 | 1.528 | 0.228 | 4.504 | 0.016 | 8.477 | 2 | 0.014 |
| Self-Regulation Strategies During Learning | 1.609 | 0.211 | 1.748 | 0.185 | 2.834 | 0.069 | 5.799 | 2 | 0.055 |
| Self-Regulation Strategies After Learning | 0.539 | 0.587 | 0.494 | 0.614 | 1.129 | 0.332 | 2.000 | 2 | 0.368 |
| Time Management | 1.363 | 0.266 | 0.576 | 0.566 | 0.091 | 0.914 | 0.389 | 2 | 0.823 |
| Environment Structure | 0.006 | 0.994 | 0.025 | 0.975 | 1.460 | 0.243 | 2.215 | 2 | 0.330 |
| Regulation Persistence | 3.599 | 0.035 | 3.533 | 0.037 | 2.105 | 0.133 | 3.725 | 2 | 0.155 |
| Peer Learning | 0.227 | 0.798 | 0.124 | 0.884 | 0.563 | 0.573 | 0.544 | 2 | 0.762 |
| Help Seeking | 1.636 | 0.206 | 1.252 | 0.296 | 1.978 | 0.150 | 3.795 | 2 | 0.150 |

### 4.2.3. Analysis Item by Item

In the two constructs where significant differences in the mean values across the years were found (*Self-regulation Strategies Before learning* and *SRS During Learning)*, we analysed each item more closely. All items increased their mean values from 2019/20 to 2020/21 and decreased in the following year (Figures 4 and 5). However, the significant differences are mainly in the three following items (Table 8 Kruskal–Wallis Test, $p = 0.034$, $p = 0.006$, and $p = 0.068$), and, in the first year only, (Mann–Whitney Test, $p = 0.013$, $p = 0.001$, and $p = 0.022$, respectively):

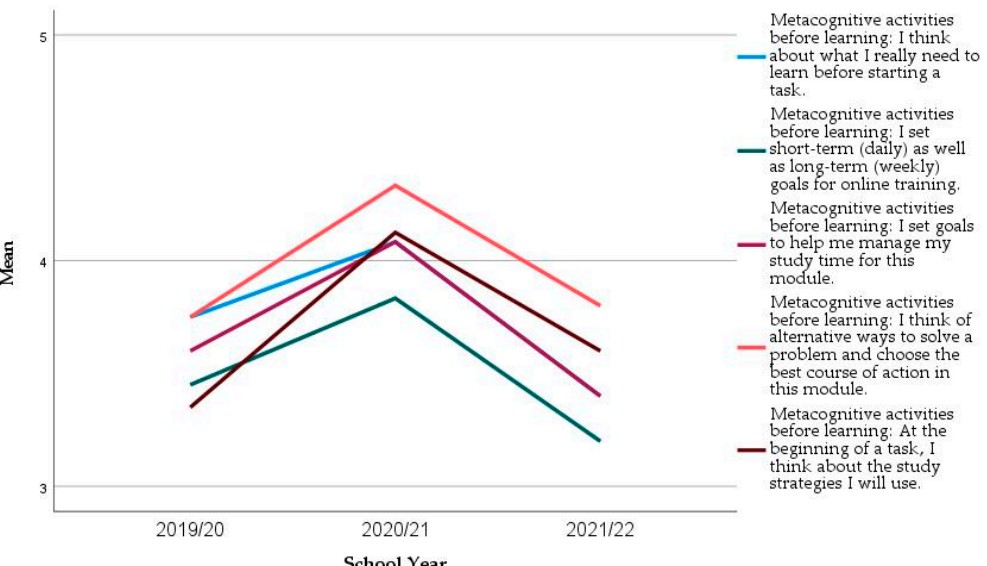

**Figure 4.** Evolution of the mean of the items from *Self-regulation activities before learning* in the longitudinal sample.

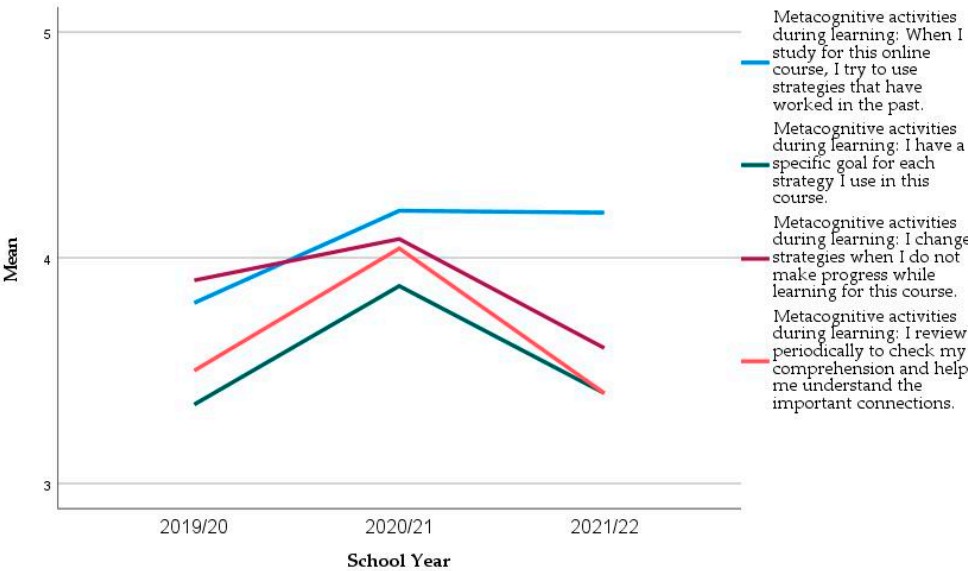

**Figure 5.** Evolution of the mean of the items from *Self-regulation activities during learning* in the longitudinal sample.

- Metacognitive activities before learning: I think of alternative ways to solve a problem and choose the best course of action in this module;
- Metacognitive activities before learning: At the beginning of a task, I think about the study strategies I will use;

- Metacognitive activities during learning: I have a specific goal for each strategy I use in this course.

**Table 8.** Kruskal–Wallis test for some of the items across the school years.

| Kruskal–Wallis Test | | | |
|---|---|---|---|
| | Kruskal–Wallis H | df | Asymp. Sig. |
| Metacognitive activities before learning: I think about what I really need to learn before starting a task. | 2.659 | 2 | 2.659 |
| Metacognitive activities before learning: I set short-term (daily) as well as long-term (weekly) goals for online training. | 1.323 | 2 | 0.516 |
| Metacognitive activities before learning: I set goals to help me manage my study time for this module. | 2.486 | 2 | 0.289 |
| Metacognitive activities before learning: I think of alternative ways to solve a problem and choose the best course of action in this module. | 6.772 | 2 | 0.034 |
| Metacognitive activities before learning: At the beginning of a task, I think about the study strategies I will use. | 10.304 | 2 | 0.006 |
| Metacognitive activities during learning: When I study for this online course, I try to use strategies that have worked in the past. | 1.488 | 2 | 0.475 |
| Metacognitive activities during learning: I have a specific goal for each strategy I use in this course. | 5.366 | 2 | 0.068 |
| Metacognitive activities during learning: I change strategies when I do not make progress while learning for this course. | 2.017 | 2 | 0.365 |
| Metacognitive activities during learning: I review periodically to check my comprehension and help me understand the important connections. | 4.511 | 2 | 0.105 |

Grouping Variable: School Year.

### 4.3. Comparing Data before and after COVID-19

As a curiosity, the general results obtained by Zarouk et al. were compared with the students enrolled in 2018–2019, before the pandemic breakout, with the results from our study carried out during and after the lockdown (in the school years 2019–20, 2020–21, and 2021–22). The Z-test was used to compare the mean values in the nine constructs. The results in Table 9 show that all constructs presented significant differences ($p$-value < 5%). The mean of all constructs increased significantly.

**Table 9.** Comparison of the results before COVID-19 and after COVID-19.

| Constructs | Before COVID-19: School Year 2018/2019 (Zarouk et al., 2020 [4]) | | | After COVID-19: School Years 2019/20, 2020/21, and 2021/22 | | | Z-Test for Comparing Means | |
|---|---|---|---|---|---|---|---|---|
| | N | Mean | Std. Deviation | N | Mean | Std. Deviation | Z | $p$-Value |
| Motivational Beliefs | 84 | 2.67 | 0.68 | 49 | 4.316 | 0.435 | −17.017 | <0.0001 |
| Self-Regulation Strategies Before Learning | 84 | 2.56 | 0.60 | 49 | 3.820 | 0.667 | −10.898 | <0.0001 |
| Self-Regulation Strategies During Learning | 84 | 2.85 | 0.51 | 49 | 3.842 | 0.629 | −9.389 | <0.0001 |
| Self-Regulation Strategies After Learning | 84 | 2.62 | 0.59 | 49 | 3.806 | 0.713 | −9.843 | <0.0001 |
| Time Management | 84 | 2.48 | 0.46 | 49 | 3.541 | 0.596 | −10.735 | <0.0001 |
| Environment Structure | 84 | 2.67 | 0.51 | 49 | 4.128 | 0.675 | −13.087 | <0.0001 |
| Regulation Persistence | 84 | 2.68 | 0.57 | 49 | 3.922 | 0.631 | −11.338 | <0.0001 |
| Peer Learning | 84 | 1.89 | 0.62 | 49 | 4.204 | 0.547 | −22.393 | <0.0001 |
| Help Seeking | 84 | 2.24 | 0.56 | 49 | 4.255 | 0.580 | −19.564 | <0.0001 |

## 5. Conclusions

In this study, we revisited the affordances of the implementation of PBL in SRL before, during and after the COVID-19 pandemic. PBL places the student at the centre of the learning process, and, as a result, self-regulation becomes essential, as it is crucial to analyse the market/situation and adapt to it accordingly. With the need to change into ERT, the relevance and significance of SRL were questioned and the necessity to analyse and assess its suitability emerged.

In the context of the Technical Translation courses at ISCAP where PBL is being used, data was collected in relation to student perception on their self-regulation competence and its development or lack thereof. Over the course of three school years (2019/2020, 2020/2021, 2021/2022), a total of six group samples were collected, three before and three after PBL implementation. Thus, a statistical analysis was performed, and the quantitative data was analysed in order to see, first, whether or not online PBL fosters the development of essential competencies for future translators, namely self-regulation. Additionally, we wanted to determine whether or not it is possible to establish a correlation between the lockdown and student SRL.

In the first phase of this study, with ERT in place, no significant differences were found regarding the impact of the Motivational Beliefs and Self-Regulation Strategies reported in the pre-test and the post-test. However, a more detailed observation showed that a higher level of self confidence in autonomous learning was achieved, but a lower level of the importance and usefulness of the course contents was noted.

Subsequently, the longitudinal analysis revealed that, with the exception of time-management, which has also been widely discussed in the literature, student self-regulation strategies increased.

In the translation labour market, the competencies needed are vast and ever-changing. Given the fluidity of the market, self-regulation becomes essential. Thus, translation training should reflect these market needs for success. PBL approaches enable the development of competencies, self-regulation included, and simulate the translator's working environment. Thus, PBL is seen as a positive methodology, which enhances SRL and other competencies development in translation training. COVID-19 and the consequent lockdown may have led to ERT, but the work implemented, such as course design and student-centred teaching-and-learning approaches, for example, have proven to be beneficial for the development of transversal competencies and skills, becoming established practices in educational institutions.

The study described is not without its limitations. One is the fact that it is a quasi-experiment, and, in itself, the design raises issues of comparability and the rationale between cause and effect. Additionally, participation, although highly recommended, was not mandatory. For that reason, students frequently choose not to respond, as was shown in the final year of the study, where there were only five respondents. Therefore, it is not possible to use the data from 2021–2022 confidently.

Student participation was voluntary. However, they were informed that surveys would be coded for pre- and post-analyses. We may reflect on student understanding of confidentiality as another issue, which may have, in some way, influenced student participation. Even though it is crucial to identify participants in order to establish correlations between the pre-test and the post-test, it may be possible that students would answer questions in a way that they believe the teacher is anticipating.

The impact and effectiveness of an online-PBL approach on student SRL is satisfactory, and we suggest that future studies readdress other constructs within this competence, possibly followed with in-depth interviews, to understand the student rationale behind some of the findings. To address the lack of participation, a possible solution might be the distancing of the lecturers from the actual research.

**Author Contributions:** Conceptualization, C.T., S.R. and G.C.; methodology, S.R.; validation, C.T., S.R., C.L. and G.C.; formal analysis, C.L.; investigation, S.R. and G.C.; resources, C.T., S.R., C.L. and G.C.; data curation, C.L.; writing—original draft preparation, C.T.; writing—review and editing, C.T., S.R., C.L. and G.C.; visualization, C.T. All authors have read and agreed to the published version of the manuscript.

**Funding:** This research received no external funding.

**Institutional Review Board Statement:** Not applicable.

**Informed Consent Statement:** The study was conducted in accordance with the Declaration of Helsinki and approved by the Dean of the Institution. Informed consent was obtained from all subjects involved in the study.

**Data Availability Statement:** Data is contained within the article.

**Conflicts of Interest:** The authors declare no conflict of interest.

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
