# Peer review of "Competence Development Strategies after COVID-19: Using PBL in Translation Courses"

_education, doi:10.3390/educsci13030283_

Round 1

Reviewer 1 Report

Thank you for your submission. The premise and logic of your paper are of interest to the field, but because of the very poor paper organization (unclear introduction, sweeping grand statements without attribution, and an inconsistent paper focus/ outlining) makes this paper unpublishable at this stage. I encourage you to do a major rework by first organizing the paper with clear orientations for the reader and then from this unpack key sections in a clear and connected manner. It really is not until Lines 193-194 that I got real clarity on what you were on about.

Author Response

Thank you for considering that the premise and logic of our paper is of interest to the field.

The paper has been reorganized; the introduction is now clearer, and the outline of the paper has been clarified.

The document has been completely revised, in terms of language and structure.

We welcome any further comment which may help us improve the article.

Reviewer 2 Report

See the document attached.

Author Response

Answers to Reviewers of MDPI Education Sciences – Round 1

Paper title: Competence development strategies after Covid: using PBL in translation courses

Reviewer 2

Thank you very much for your comments and suggestions.

The table below lists the comments and our responses.

Comments/Suggestions

Answers/Revisions

1

Maybe the Introduction (contextualization) in this abstract is so long. And also it could be interesting to explain about methodology, and some results and conclusions as well.

The abstract has been rewritten and now contains more details about the methodology and results.

2

How does the PBL work to strengthen self-regulation or self-regulated learning?

We believe this has been addressed in the new abstract.

3

Missing to specify the page number.

added

4

in the specific topic of self-regulated learning

added

5

It could be interesting to know some different cases planned in PBL

Section 2.3 addresses some other cases.

"Studies on PBL in translation training are not new (González and Díaz [22], Li, Zhang and He ’[23], Moghaddas and Khoshsaligheh [24], and Apandi and Afiah [25])."

6

Do we have some data about self-regulated learning's students from before pandemics?

Experimental study implemented in 2018-2019, prior to the pandemic, led by Mohamed Yassine Zarouk is published in [4].

This is reinforced in the Introduction and Methods section.

The results are compared with our data in Table 9.

7

Specify in which dimensions of SRL.

The three pillars we analyzed are detailed in the instrument and results.

Should this be made clearer also in this section?

8

specify: (PBL hereinafter)

Corrected

9

Did we do a pre-test and post-test? (before vs after COVID19)

In our quantitative quasi-experimental study (non-random, to study possible cause/effect of PBL on student perception of self-regulated learning) the pre-test is the survey (questionnaire) answered by students every year in the beginning of the course, and the post-test is the survey answered by the same students at the end of the course. The teaching and learning method of the course is PBL. Every year there is a different set of students. Therefore, each student answered both the pre-test and post-test. The results before COVID are published in [4]. Our data corresponds to the Covid years, but we also cross analyze our data with the results before covid.

We used the concept "longitudinal study" because we observe our students over a period of six months and compare each student's survey results. However, we also analyze the all the results over the years, i.e.,   every year we analyze a new group of students.

We appreciate any suggestion you may have.

10

Missing to specify the page number.                                 added

11

By what criteria is it stipulated?

An analysis of the 36 skill descriptors, organized by competence area, shows that several descriptors may indeed be related to self-regulated learning. We decided to identify this particular descriptor as an example. This is explained in the in the new version, lines 99-101

12

Mention can also be made of the soft skills of 21st century education

This was added, line 101

13

self-regulated learning

The correction was made

14

It is necessary to unify the way of writing. Sometimes some letters are capitalized, sometimes others. Unify writing style.

The correction was made

15

This should be Michel et al. (because there are 3 authors).

This and similar corrections were made.

16

It suggested to write in a more formal and impersonal form.

The correction was made

17

¿before lockdown - COVID19?

No, the pre-test designation corresponds to before PBL and the post-test corresponds to after PBL.  The pre tests and post tests happened every year, both in the years before covid and also in the years after covid.

18

How many students are they?

In line 214-216: the total number of students enrolled: 91 students in 2019/20, 82 in 2020/21 and 79 in 2021/22. Consistency was only found in 2020/2021, with 16 participants (valid surveys). This is why we decided to look at the data of the different years.

19

It remains to apply methodology with regard to:

Time / period of data collection

Ethical aspects of research, e.g. Informed consent

Data analysis: types of analysis, statistics...

Methods section was rewritten.

Data collection: lines 208-209

Mention to the informed consent and the type of statistical analysis was added.

20

“no significant differences”

The sentence was rewritten

21

Maybe it is no necessary to write them in bold.

Completed

22

Be careful: above this value (0.9) it is considered that there is redundancy or duplication

The full questionnaire registered Alpha=0.933 and the "Help seeking" construct 0.919, but in the remaining dimensions and constructs it is not higher than 0.891. Also, "Cronbach alpha values if item is deleted" were analyzed and did no point out to removing critical items, except Time Management. Extreme correlation coefficients between total and items were not detected. The questionnaire was developed by other authors, and mainly for comparability reasons we opted not to exclude any of the original questions.

23

¿3 academic courses with the same sample? Specify. ¿Or maybe it is different sample?

Every year there is a different set of students undertaking this one course taught with PBL strategies. The sentence was rewritten.

24

ERL specify

It was corrected to ERT, as defined in line 44

25

Review references according to journal guidelines

Completed

Round 2

Reviewer 1 Report

The revised paper is a significant improvement on the first. The level of English use and the resulting narrative makes for easy reading and the study logic is now much clearer. The outlining of the study/ data is sound, however, I would have liked to see the RQs used more in the paper, especially to frame statements made in the conclusion. Put simply it is the RQ that drives this study and their outlining in the introduction focuses the reader, and thus I was wanting to have them clearly articulated as finding in the conclusion. With a more focused approach to answering the RQs in statements in the conclusion, you will increase/ define the contribution of the paper. Dealing with them also is the 'test' that the paper achieves what you've said it seeks to do. Meaning,  focus on what's the contribution to the field?

Author Response

Dear Reviewer 1.

Thank you very much for your contribution to this paper. Your feedback and comments have, indeed, enriched the manuscript considerably.

In this second review, we added, as suggested, the RQs to guide the reader and help maintain the focus: 

- lines 254-257: at the beginning of section 4;

- lines 452-455: in the conclusion of the paper.

Please let us know if we should address any other issues.

Sincerely,

Sandra

Reviewer 2 Report

Dear authors, thank you for your changes.

Author Response

Dear Reviewer 2.

In this second review, we added, as suggested by Reviewer 1, the RQs to guide the reader and help maintain the focus: 

- lines 254-257: at the beginning of section 4;

- lines 452-455: in the conclusion of the paper.

Please let us know if we should address any other issues.

Sincerely,

Sandra
